# Endometrial Epithelial ARID1A Is Required for Uterine Immune Homeostasis during Early Pregnancy

**DOI:** 10.3390/ijms23116067

**Published:** 2022-05-28

**Authors:** Ryan M. Marquardt, Soo Hyun Ahn, Jake J. Reske, Ronald L. Chandler, Margaret G. Petroff, Tae Hoon Kim, Jae-Wook Jeong

**Affiliations:** 1Department of Obstetrics, Gynecology and Reproductive Biology, Michigan State University, Grand Rapids, MI 49503, USA; marqua45@msu.edu (R.M.M.); reskejak@msu.edu (J.J.R.); rlc@msu.edu (R.L.C.); kimtae31@msu.edu (T.H.K.); 2Cell and Molecular Biology Program, College of Natural Science, Michigan State University, East Lansing, MI 48824, USA; petrof10@msu.edu; 3Department of Pathobiology and Diagnostic Investigation, College of Veterinary Medicine, Michigan State University, East Lansing, MI 48824, USA; ahnsooh1@cvm.msu.edu; 4Department of Microbiology and Molecular Genetics, Michigan State University, East Lansing, MI 48824, USA

**Keywords:** endometrium, infertility, immunology, inflammation, endometriosis, ARID1A

## Abstract

A growing body of work suggests epigenetic dysregulation contributes to endometriosis pathophysiology and female infertility. The chromatin remodeling complex subunit AT-rich interaction domain 1A (ARID1A) must be properly expressed to maintain normal uterine function. Endometrial epithelial ARID1A is indispensable for pregnancy establishment in mice through regulation of endometrial gland function; however, ARID1A expression is decreased in infertile women with endometriosis. We hypothesized that ARID1A performs critical operations in the endometrial epithelium necessary for fertility besides maintaining gland function. To identify alterations in uterine gene expression resulting from loss of epithelial ARID1A, we performed RNA-sequencing analysis on pre-implantation uteri from *Ltf^iCre/+^Arid1a^f/f^* and control mice. Differential expression analysis identified 4181 differentially expressed genes enriched for immune-related ingenuity canonical pathways including agranulocyte adhesion and diapedesis and natural killer cell signaling. RT-qPCR confirmed an increase in pro-inflammatory cytokine and macrophage-related gene expression but a decrease in natural killer cell signaling. Immunostaining confirmed a uterus-specific increase in macrophage infiltration. Flow cytometry delineated an increase in inflammatory macrophages and a decrease in uterine dendritic cells in *Ltf^iCre/+^Arid1a^f/f^* uteri. These findings demonstrate a role for endometrial epithelial ARID1A in suppressing inflammation and maintaining uterine immune homeostasis, which are required for successful pregnancy and gynecological health.

## 1. Introduction

The endometrium, the inner lining of the uterus, is a highly dynamic and interconnected group of cells consisting of luminal epithelium, glandular epithelium, and stroma. Luminal epithelial cells face the uterine lumen and are the first uterine cells to interact with an embryo in the context of pregnancy. Glandular epithelial cells form gland structures apart from the uterine lumen and perform critical functions for pregnancy through their secretions, which contact the contents of the lumen, the luminal epithelium, and the surrounding stroma [1]. The stroma plays a supportive role from its position surrounding the epithelial cells and consists of a mixture of stromal fibroblasts, endothelial cells, and immune cells.

Together, these endometrial compartments maintain a dynamic homeostasis through tight regulation by the ovarian steroid hormones estrogen (E2) and progesterone (P4). Signaling primarily through their nuclear receptors, E2 and P4 function in a complex epithelial-stromal crosstalk that governs the menstrual cycle in humans, the estrous cycle in mice, and the window of implantation in both species [2,3]. Additionally, the differentiation of uterine stromal fibroblasts into specialized, secretory decidual cells is essential for pregnancy progression and is a progesterone-dependent cyclical process in humans, whereas it relies on the presence of an embryo and nidatory E2 to induce leukemia inhibitory factor (LIF) secretion from glands in rodents [4,5]. Proper embryo implantation and complete decidualization are required not only for the establishment of pregnancy but also for the healthy progression of pregnancy to placentation and on-time delivery [6].

To facilitate proper vascularization and tissue remodeling in the formation of the decidua, which goes on to act as a physical scaffold and provide essential nutritional support and immune tolerance for the developing embryo until placentation, at least three specific immune cell types must maintain an appropriate spaciotemporal balance [7]. Uterine natural killer cells (uNKs), an innate lymphoid cell type, constitute the majority of all human decidual lymphocytes and contribute to angiogenesis, vascular remodeling, and proper placentation [7,8]. More recently, a critical role for uNKs has been described in clearing senescent decidual cells [9,10]. In mice, uNKs appear by gestation day (GD) 6.5, just after implantation, peak at GD 12.5 after the placenta is fully formed, and play a role in decidual integrity [11].

Macrophages, a class of innate myeloid immune cells, are another common cell type in the uterus, and they can originate from circulating monocytes or be resident to the tissue [7]. These cells are key sensors of infection and regulators of the inflammatory response, and subtypes of macrophages can be either proinflammatory or anti-inflammatory depending on the cytokines they secrete. Homeostatic uterine macrophages are thought to be primarily anti-inflammatory outside of transient inflammation in preparation for pregnancy, but an abundance of proinflammatory macrophage activity in the uterus is associated with pathologies [12].

Dendritic cells in general are myeloid antigen-presenting cells that link innate immunity to the adaptive immune system. Uterine dendritic cells (uDCs) are normally rare, but increasing numbers are recruited to the decidua and appear to function there in tissue remodeling and angiogenesis [7,13,14]. Numbers of uNKs, uterine macrophages, and uDCs have all been reported to fluctuate during endometrial cycling and peak at the secretory phase, implying either direct or indirect regulation by steroid hormones, but the mechanism has not been well characterized [15].

When the normal hormone and immune-regulated homeostatic balance of the endometrium is lost, gynecological pathologies can develop including recurrent implantation failure, recurrent pregnancy loss, endometrial cancer, and endometriosis [16]. Endometriosis occurs when endometrium-like glands and stroma form ectopic lesions outside the uterus, and along with a high prevalence of chronic pain, it is often accompanied by infertility or subfertility [17]. The prevalence of this E2-driven, P4-resistant inflammatory disorder is estimated to be 10% of reproductive-age women, but the heterogeneity of its clinical presentation contributes to a high frequency of delayed diagnosis, making true prevalence estimates and systematic study difficult [18]. Surgical resection of lesions and hormone suppression are the standard treatment options for endometriosis-related pain, but suppression of ovulation precludes fertility, and surgical treatments have mixed results on fertility outcomes, leaving in vitro fertilization as the typical option for women affected by endometriosis-related infertility [19]. While its pathogenesis is not thoroughly understood and is almost certainly multifactorial, perturbations in local and systemic immune cell populations have been clearly implicated in both endometriotic lesion establishment and dysregulation of the endometrial environment [15].

Substantial work from our group and others has also identified a wide array of epigenetic factors dysregulated in the endometrium of women with endometriosis including histone-modifying enzymes, DNA modifiers, and chromatin architecture modifiers [20,21,22,23,24,25]. The SWItch/Sucrose Non-Fermentable (SWI/SNF) chromatin remodeling complex subunit AT-rich interaction domain 1A (ARID1A) is commonly known for its tumor suppressor role and its prevalent inactivating mutations in endometriosis-associated ovarian carcinomas, but it is also mutated in non-cancerous endometriotic lesions and underexpressed in the endometrium of infertile women with endometriosis [24,26,27]. Furthermore, studies using mouse models have shown that deletion of uterine *Arid1a* drives increased endometriosis-like lesion establishment and causes endometrial-factor infertility related to disrupted P4 and E2 signaling [22,24,28]. Focused study on the role of ARID1A in the endometrial epithelium has revealed its critical cell-type-specific roles of maintaining epithelial identity and enabling gland development and function in pregnancy [29,30,31]. Deletion of *Arid1a* in the adult mouse endometrial epithelium led to early pregnancy defects through attenuation of forkhead box A2 (FOXA2) expression and LIF secretion from uterine glands. However, disruption of this pathway does not appear to fully explain the pregnancy-related uterine defects resulting from epithelial *Arid1a* deletion, which led us to hypothesize that endometrial epithelial ARID1A performs other critical functions in the endometrial epithelium necessary for fertility.

In this study, we utilized RNA-sequencing analysis of endometrial epithelium-specific *Arid1a* knockout mice driven by lactoferrin (*Ltf*)-*iCre* (*Ltf^iCre/+^Arid1a^f/f^*) to explore the in vivo uterine transcriptome dysregulation in early pregnancy that results from loss of epithelial ARID1A. Our analysis revealed large scale disruption of immune-related pathways, most notably an increase in proinflammatory cytokine gene expression and a decrease in uNK cell markers. Further functional study demonstrated a marked increase in uterine infiltration by proinflammatory macrophages with correspondent decreases in uNKs and uDCs, revealing a critical role for endometrial epithelial ARID1A in maintaining uterine immune homeostasis during early pregnancy.

## 2. Results

### 2.1. Deletion of Endometrial Epithelial Arid1a in Mice Causes Diminished Implantation Site Size and uNK Cell Numbers at GD 7.5

*Ltf^iCre/+^Arid1a^f/f^* mice are severely subfertile and exhibit implantation and decidualization defects at the peri-implantation and early post-implantation stages (GD 4.5–5.5), but the effects of endometrial epithelial *Arid1a* deletion on subsequent stages of pregnancy have not been previously reported [31]. We collected uterine samples at GD 7.5 and found that although the number of implantation sites was not different from controls (Control = 4.88 ± 1.47, *Ltf^iCre/+^Arid1a^f/f^* = 5.00 ± 1.13, *p* = 0.8578), *Ltf^iCre/+^Arid1a^f/f^* implantation sites were significantly decreased in diameter (Control = 3.80 ± 0.05 mm, *Ltf^iCre/+^Arid1a^f/f^* = 2.89 ± 0.13 mm, *p* = 0.0002) and weight (Control = 0.0092 ± 0.0006 g, *Ltf^iCre/+^Arid1a^f/f^* = 0.0060 ± 0.0004 g, *p* = 0.0092; Figure 1).

Immunohistochemical staining of implantation site cross sections for the decidualization marker cyclooxygenase-2 (COX-2) revealed a substantially diminished decidual area (Figure 2A). Concurrent with the development of the decidua, uNK cells normally proliferate in a healthy implantation site and function in angiogenesis and vascular remodeling [7,11]. However, *Ltf^iCre/+^Arid1a^f/f^* mice exhibited significantly decreased numbers of uNK cells at GD 7.5 counted based on *Dolichos biflorus* (DBA) lectin staining (Control = 390.00 ± 90.63, *Ltf^iCre/+^Arid1a^f/f^* = 13.40 ± 5.92, *p* = 0.0079; Figure 2B,C).

### 2.2. RNA-Sequencing Analysis of Ltf^iCre/+^Arid1a^f/f^ Uteri at GD 3.5 Reveals Altered Immune Pathways

Seeking to discover gene expression changes earlier in pregnancy that could explain compromised decidualization and deficiency of uNK cells due to *Arid1a* deletion, we performed RNA-sequencing analysis on *Ltf^iCre/+^Arid1a^f/f^* uteri collected at GD 3.5, when the uterus is preparing to receive an implanting embryo and just before the onset of decidualization. Principal component analysis of filtered, normalized log-transformed gene counts showed that the five *Ltf^iCre/+^Arid1a^f/f^* and five control *Arid1a^f/f^* samples segregated from one another based on overall gene expression (Figure 3A). Differential expression analysis revealed that 4181 uterine genes (2174 increased, 2007 decreased) were significantly dysregulated due to deletion of endometrial epithelial *Arid1a*, which is visualized by hierarchical clustering heatmap (FDR < 0.05; Figure 3B; Appendix A). Ingenuity Pathway Analysis (IPA) of this gene set identified the most statistically significant overlaps with canonical pathways related to lipid biosynthesis, cell cycle regulation, and immune function and with upstream regulators including steroid hormones, cytokines, and other immune modulators (Figure 3C,D).

Because of the large number and broad spectrum of immune-related pathways in the dataset and the known change in uNK cells in *Ltf^iCre/+^Arid1a^f/f^* uteri, we examined the immune-related changes more closely. Altered immune-related canonical pathways involved both the innate and adaptive immune response and both lymphoid and myeloid immune cells (Figure 4A). Upstream regulator analysis revealed activation of inflammatory factor targets, notably interferon gamma (IFNG), tumor necrosis factor (TNF), and interleukin-1 beta (IL1B), and deactivation of target molecules for the anti-inflammatory interleukin-10 (IL-10) receptor (IL10RA; Figure 4B).

RT-qPCR validation of individual differentially expressed genes from the RNA-seq data confirmed that several genes coding for proinflammatory cytokines (interleukin-36 alpha, *Il36a*; interleukin-17A, *Il17a*; colony-stimulating factor 2, *Csf2*; interleukin-1 alpha, *Il1a*; *Tnf*; interleukin-18, *Il18*; colony-stimulating factor 3, *Csf3*; tumor necrosis factor ligand superfamily member 13B, *Tnfsf13b*) and cytokine receptors (C–C motif chemokine receptor 2, *Ccr2*; C–C motif chemokine receptor 4, *Ccr4*) were highly upregulated in the *Ltf^iCre/+^Arid1a^f/f^* uterus (*p* = 0.0079, 0.0079, 0.0079, 0.0079, 0.0159, 0.0001, 0.0013, 0.0469, 0.0079, 0.0317, respectively; Figure 4C). One receptor component (interleukin-17 receptor B, *Il17rb*) was decreased, possibly a compensatory effect due to the incredibly high levels of *Il17a* (*p* = 0.0317; Figure 4C). Genes related to innate immunity were also modulated, with one inflammasome-related gene (NLR family, apoptosis inhibitory protein 1, *Naip1*) increased and another (myeloid differentiation primary response 88, *Myd88*) decreased (*p* = 0.0079, 0.0013, respectively; Figure 4C).

The NK cell receptor gene (killer cell lectin-like receptor 7, *Klra7*) was downregulated, consistent with the finding of diminished uNK cells at GD 7.5 (*p* = 0.0316; Figure 4C). Scavenger receptor class A member 5 (*SCARA5*) and iodothyronine deiodinase 2 (*DIO2*) have been identified as marker genes for a diverging decidual response in vivo [10]. A shift toward dominance of senescent decidual cells, indicated by decreased *SCARA5* and increased *DIO2*, is associated uNK cell-deficient pre-pregnancy endometrium in humans [10]. Along with decreased uNK cell signaling and cell proliferation, *Ltf^iCre/+^Arid1a^f/f^* uteri exhibited significantly decreased *Scara5* and increased *Dio2* expression, though not quite to a level of statistical significance (*p* = 0.0480, 0.0952, respectively; Figure 4C). Together, these results indicate that a highly proinflammatory uterine environment results from deletion of endometrial epithelial *Arid1a* concomitant with a decrease in normal uNK cell regulation of the endometrium.

### 2.3. Uterine Macrophage Numbers Are Elevated in Ltf^iCre/+^Arid1a^f/f^ Mice at GD 3.5

Next, we asked if the pro-inflammatory gene expression in the *Ltf^iCre/+^Arid1a^f/f^* uterus at GD 3.5 was intrinsic to the endometrial stromal and epithelial cells or if it coincided with altered immune cell populations. Since macrophages are one of the major cell types that produce proinflammatory cytokines, such as TNF and IL-1A, and CSF2 and CCR2 are involved in the proliferation and homing of monocytes and macrophages, we assessed uterine macrophages in *Ltf^iCre/+^Arid1a^f/f^* mice by immunostaining for EGF-like module-containing mucin-like hormone receptor-like 1 (F4/80), a cell surface marker for macrophages and, to a lesser extent, monocytes [32,33]. While F4/80 positive cells were common in the control uterus, comprising approximately 30% of the stromal compartment, the percent of F4/80 positive cells was significantly increased, nearly doubling, in *Ltf^iCre/+^Arid1a^f/f^* uteri (Control = 27.86 ± 5.03, *Ltf^iCre/+^Arid1a^f/f^* = 57.30 ± 5.49, *p* = 0.0042; Figure 5). To determine if the changes in macrophage numbers were systemic or specific to the uterus, we stained control and *Ltf^iCre/+^Arid1a^f/f^* spleen tissue for F4/80, finding no difference in macrophage appearance or number (Control = 93.73 ± 1.43, *Ltf^iCre/+^Arid1a^f/f^* = 93.55 ± 1.80, *p* > 0.9999, Appendix A).

To better characterize and more comprehensively assess the myeloid immune cell population in the *Ltf^iCre/+^Arid1a^f/f^* GD 3.5 uterus, we performed flow cytometry analysis, selecting among live, singlet leukocytes for a cluster of differentiation molecule 11B (CD11b)+ major histocompatibility complex class II (MHCII)+ cells, which were not different between genotypes (*p* = 0.5396, Figure 6A, Appendix A). Within this myeloid population, F4/80− cluster of differentiation 64 (CD64)− lymphocyte antigen 6C (Ly6C)+ cells were unchanged, F4/80+ CD64+ Ly6C+ cells were significantly increased, and F4/80+ CD64+ Ly6C− cells were significantly decreased (*p* = 0.8154, 0.0050, 0.0029, respectively; Figure 6A,B, Appendix A). Since CD64 is expressed primarily on macrophages, and Ly6C is thought to be specific to circulation-derived inflammatory monocytes and macrophages, the increase in F4/80+ macrophages in *Ltf^iCre/+^Arid1a^f/f^* uteri appears to be driven by infiltration of circulating Ly6C+ inflammatory cells rather than new growth of resident macrophages [33,34,35,36,37,38]. Further myeloid immune cell analysis revealed that Ly6G+ neutrophil numbers were consistent between genotypes, but CD64− cluster of differentiation 24 (CD24)+ uDCs, important for proper decidua formation, were significantly decreased in *Ltf^iCre/+^Arid1a^f/f^* mice [13,38] (*p* = 0.7818, 0.0016, respectively; Figure 6C,D, Appendix A).

To assess the extrauterine peritoneal myeloid immune environment, we quantified monocytes, CD11b+ cells, large peritoneal macrophages, and small peritoneal macrophages in matched peritoneal fluid [39]. This analysis yielded no significantly altered peritoneal immune cell populations, suggesting again that the changes observed in myeloid immune cell composition in GD 3.5 *Ltf^iCre/+^Arid1a^f/f^* mice are specific to the uterus (*p* = 0.3214, 0.5941, 0.6783, 0.7649, respectively; Appendix A). Together, these data demonstrate that deletion of endometrial epithelial *Arid1a* causes uterine-specific infiltration of circulating inflammatory myeloid cells during early pregnancy, resulting in elevated numbers of inflammatory macrophages alongside decreased uDCs and altered uNK cell signaling in the uterus.

## 3. Discussion

In gynecological pathologies such as endometriosis and infertility, the normally tightly regulated epigenetic landscape and immune environment of the endometrium are thrown off balance [40]. We previously showed that endometrial ARID1A levels are diminished in women with endometriosis and that abolishing endometrial epithelial ARID1A expression drives a loss of endometrial receptivity and failure of pregnancy establishment and maintenance [24,31]. Past research has also demonstrated that the endometrium of women with endometriosis experiences an increase in proinflammatory macrophages and alterations in uNK cell and uDC activity [15]. In this study, we describe a possible mechanism for a connection between these phenomena.

When we deleted *Arid1a* the endometrial epithelium using *Ltf^iCre/+^Arid1a^f/f^* mice, massive changes in immune-related gene expression patterns resulted in the pre-implantation (GD 3.5) uterus. Uterine expression of the proinflammatory cytokine genes *Il36a*, *Il17a*, *Csf2*, *Il1a*, *Tnf*, *Il18*, *Csf3*, and *Tfnfs13b* spiked alongside increased uterus-specific infiltration of F4/80+ CD64+ Ly6C+ inflammatory macrophages. At the same time, uDC numbers were diminished and pro-decidual uNK markers decreased. These disruptions were followed by reduced implantation site size, compromised decidua formation, and dramatic diminution of uNK cell presence post-implantation (GD 7.5). Interestingly, increases in *IL-17A*, *IL18*, *TNFSF13B*, and TNF signaling have all been previously identified in endometriosis conditions [41,42,43]. Increased IL-18 has been linked to failed embryo transfer and recurrent miscarriage [44,45]. Furthermore, TNF can inhibit trophoblast invasion, and one report showed that inhibiting its activity may improve live birth rates in women with recurrent spontaneous abortion [46,47]. Finally, high levels of uterine IL-36A were shown to correlate with an increased rate of fetal loss in mice [48].

Whether uNK cell numbers are altered in women with endometriosis is controversial; however, they do appear to be phenotypically altered with increased cytotoxicity [15]. In human pregnancy, the literature is clear regarding the importance of uNK cells for vasculature remodeling in the formation of the decidua and later on the placenta [49]. Moreover, a role for uNK cells in clearing senescent decidual cells has recently been described as important for maintaining uterine homeostasis and implantation [9]. An appropriate spaciotemporal balance between *SCARA5*+ decidual cells and *DIO2*+ senescent decidual cells must be maintained to allow implantation and prevent pregnancy loss, and low *SCARA5*, high *DIO2*, and low uNK populations are associated with recurrent pregnancy loss [10]. Our data from early pregnancy in *Ltf^iCre/+^Arid1a^f/f^* mice suggests that a lack of epithelial ARID1A leads to an early skew toward a pro-senescent decidual state that corresponds to the lack of sufficient uNK cells in the decidual post-implantation. Though uNK cells regulate the structure of the decidua and development of the placenta, they are not necessary in rodents for the delivery of normal numbers of pups [50,51,52]. Therefore, it is not clear if the diminished uNK cell population in *Ltf^iCre/+^Arid1a^f/f^* mice contributes to pregnancy failure or is simply a byproduct of the compromised early decidualization response. It also remains to be seen whether this phenomenon is a factor in endometriosis-related infertility in the human case.

The density of mature uDCs has been shown to be reduced in the endometrium of women with endometriosis potentially altering coordination of adaptive immunity and angiogenesis, though the functional outcome has not been thoroughly studied [15,53,54]. In mice, depletion of uDCs during pregnancy resulted in defective implantation, improper decidua formation, and embryo resorption [13]. Therefore, the reduction in uDCs following deletion of endometrial epithelial *Arid1a* in *Ltf^iCre/+^Arid1a^f/f^* mice may contribute to their compromised implantation and decidualization.

In endometriosis conditions, research to date has indicated that a decrease in anti-inflammatory macrophages and an increase in total macrophages accompanies upregulated proinflammatory cytokine signaling [15]. Similarly, our RNA-seq differentially expressed gene upstream regulator analysis identified deactivation of anti-inflammatory IL10RA target molecules but strong activation of proinflammatory TNF signaling targets in *Ltf^iCre/+^Arid1a^f/f^* uteri. TNF is considered a “master-regulator” of inflammatory cytokine production, and it can be both produced by and activate macrophages [55].

To analyze the relationship between increased inflammatory gene expression and macrophage cell presence in *Ltf^iCre/+^Arid1a^f/f^* uteri, we quantified macrophages with immunostaining and flow cytometry. Immunostaining identified a uterus-specific increase in total F4/80+ macrophages when endometrial epithelial *Arid1a* was deleted, but this analysis could not determine the source or characteristics of the cells since F4/80 marks tissue-resident macrophages as well as macrophages derived from circulating monocytes [38]. In our flow cytometry analysis, we were able to discriminate among F4/80+ CD64+ macrophages utilizing the monocyte and monocyte-derived cell marker Ly6C [33,35,37,38]. Since the Ly6C+ population increased in *Ltf^iCre/+^Arid1a^f/f^* uteri and the Ly6C− population decreased, we conclude that the increased F4/80+ macrophages in the uterus were derived from circulating inflammatory monocytes [33,35,37,38]. Since the F4/80− CD64− Ly6C+ cell populations were not different between genotypes, it appears that the altered cell type should be defined as an inflammatory monocyte-derived macrophage population rather than a classical monocyte population. However, we cannot conclude this with certainty since some monocytes also express F4/80 and CD64, and there is disagreement over whether Ly6C+ cells should be labeled as macrophages [33,35,37,38].

The observations of increased uterine proinflammatory cytokine gene expression and increased inflammatory macrophage infiltration in *Ltf^iCre/+^Arid1a^f/f^* mice raise the question of what factor first initiates the inflammatory conditions after epithelial deletion of *Arid1a*. Some level of inflammation is normal and required to facilitate implantation, but the inflammatory conditions we observed in *Ltf^iCre/+^Arid1a^f/f^* uteri were far beyond normal control levels. E2 signaling is generally considered to be proinflammatory, although this is a generalization of highly complex and context-dependent realities [56]. We previously reported a skew toward E2 signaling dominance in *Ltf^iCre/+^Arid1a^f/f^* uterine gene expression, which could contribute to the inflammatory environment, but serum E2 and P4 levels were not significantly altered [31]. We also previously demonstrated a defect of uterine gland function in *Ltf^iCre/+^Arid1a^f/f^* mice, which resulted in diminished *Lif* expression [31]. LIF regulates uterine immune cell composition in mice; however, *Lif* knockouts have half the normal numbers of uterine macrophages and double the uNKs, which is opposite from our findings here [57].

A more plausible mechanism for the triggering of extraordinary inflammation in *Ltf^iCre/+^Arid1a^f/f^* pregnant uteri is suggested by the fact that epithelial cells in the uterus as well as in other tissue types are known to produce TNF, CSF2, and other proinflammatory cytokines in certain circumstances [42,58,59,60,61]. In fact, ARID1A has been shown to directly bind near the promoter regions of *Il36a*, *Tnfsf13*, and other TNF-signaling related genes in the murine endometrial epithelium [30]. Furthermore, in 12Z endometriotic epithelial cells, genes bound by ARID1A and upregulated by knockdown of *ARID1A* expression include TNF signaling-related and inflammatory response pathway genes [29]. One gene from this ARID1A-bound and regulated group is *CCL2*, the gene for monocyte chemoattractant protein-1 (MCP-1), a major chemoattractant that regulates the migration of monocytes and macrophages by binding its receptor CCR2 [62]. Expression of both genes increases in endometriosis conditions [41,63]. Interestingly, we also found *Ccr2* gene expression increased in *Ltf^iCre/+^Arid1a^f/f^* uteri. Together, these data suggest a mechanism where loss of endometrial epithelial ARID1A leads to cell-autonomous secretion of CCL2, TNF, CSF2, and other inflammatory factors, which leads to recruitment of inflammatory macrophages into the uterus, further exacerbating the inflammatory environment (Figure 7). Such a strong inflammatory response in the uterus can then result in decreased uDCs and uNKs as we observed in *Ltf^iCre/+^Arid1a^f/f^* uteri as well as generally hostile, non-receptive conditions [33].

One limitation of our study is that all our gene expression analyses were performed on whole uterine tissue, which did not allow us to distinguish between gene expression in endometrial epithelial, stromal, and immune cells. Therefore, we cannot be certain whether the increased expression of proinflammatory cytokine genes we observed in *Ltf^iCre/+^Arid1a^f/f^* uteri was driven by direct changes in the epithelial cells or indirect changes in stromal or immune cells. Additionally, *Ltf-iCre* can reportedly be expressed in neutrophils and other myeloid lineage immune cells [64,65]. Although we did not observe changes in the number of uterine neutrophils, and systemic myeloid immune cell alterations were not evident as assessed in the spleen or peritoneal fluid, we cannot rule out the possible contribution of *Arid1a* deletion in *Ltf^iCre/+^Arid1a^f/f^* myeloid immune cells themselves. Furthermore, we did not assess whether changes occurred in uterine adaptive immune cell populations or in the uterine immune environment of non-pregnant mice, both of which are areas of interest for future study. Finally, all the experiments in this study utilized mice as a model system. While mice share many aspects of uterine biology with humans, there are important differences of reproductive tract development and function between species that must be considered [66]. For example, the differences in the hormonal environment between the human menstrual cycle and murine estrous cycle could impact the basal immune cell populations [15,67]. Therefore, our findings still need to be validated in human samples.

In summary, we have shown that deletion of endometrial epithelial *Arid1a* in mice causes large scale uterine transcriptome dysregulation during early pregnancy, including many genes related to immune function. Most notably, we found increases in proinflammatory cytokine expression and alterations in uNK cell signaling. At the cell level, increased proinflammatory cytokine transcription accompanied a uterine-specific influx of proinflammatory macrophages, a decrease in uDCs, and, after implantation, a diminished uNK cell population at implantation sites. For many of the immune-related changes we observed parallel observations from the dysregulated endometrial immune environment present in women with endometriosis, suggesting that diminished endometrial epithelial ARID1A in endometriosis conditions may contribute to the proinflammatory environment and negatively impact receptivity of the endometriosis-affected endometrium to pregnancy.

## 4. Materials and Methods

### 4.1. Mouse Models

All mouse procedures were approved by the Institutional Animal Care and Use Committee of Michigan State University. All mice were housed and bred in a designated animal care facility at Michigan State University under controlled humidity and temperature conditions and a 12 h light/dark cycle at 5 mice/cage maximum. Access to water and food (Envigo 8640 rodent diet) was ad libitum. Endometrial epithelial conditional *Arid1a* knockout mice were generated in mixed background C57BL/6, 129S6/SvEvTac, 129P2/OlaHsd, 129S1/SvImJ strain mice by initially crossing *Ltf^iCre/+^* [64] (Jackson Laboratory Stock No: 026030) males with *Arid1a^f/f^* [68] (Jackson Laboratory Stock No: 027717) females and then selecting *Ltf^iCre/+^Arid1a^f/f^* males and *Arid1a^f/f^* females from the F2 generation for continuous breeding. All experiments were performed in 12–15-week-old mature adult mice to ensure sufficient *Ltf^iCre/+^* activation. For breeding, one male mouse was normally placed into a cage with one female mouse. Occasionally, one male mouse was housed with two female mice to increase breeding success, and females were separated with their pups until weaning. After weaning at postnatal day 21–28, male and female littermates were separated and housed in groups at 5 mice/cage maximum until use in breeding or experiments. Mice of appropriate genotypes were randomly allocated to experimental groups, using littermates for comparisons when possible.

### 4.2. Mouse Procedures and Tissue Collection

Uterine samples from specific times of pregnancy were obtained by mating control or conditional *Arid1a* knockout female mice with proven wildtype male breeder mice defining morning of identification of a vaginal plug as GD 0.5. Part of each uterine sample isolated at GD 3.5 was snap-frozen and stored at −80 °C for RNA extraction. The remaining uterine tissues and paired spleen tissues were prepared for histological analysis by fixing for 6 h in 4% (*vol/vol*) paraformaldehyde (Fisher Scientific, Hampton, NH, USA, Cat. #04042-500), cryopreserving in a series of sucrose solutions increasing from 10% to 15% to 20% sucrose in Hanks’ Balanced Salt Solution (HBSS; Gibco, Grand Island, NY, USA, Cat. #14170-112), and deep freezing in Tissue-Tek optimal cutting temperature (OCT) compound (Sakura Finetek, Torrance, CA, USA, Cat. # 4583) at −80 °C. Whole uterine samples collected at GD 7.5 were photographed and weighed before fixing with 4% (*vol/vol*) paraformaldehyde and processing in a graded alcohol series for paraffin embedding. Implantation sites were identified and measured based on gross morphology with subsequent histological confirmation. Individual implantation sites were weighed after tissue processing but before embedding.

### 4.3. Histology and Immunostaining

For IHC and DBA lectin staining, fixed, paraffin-embedded tissues were cut at 5 μm, mounted on slides, deparaffinized, and rehydrated in a graded alcohol series. Immunostaining was performed by incubating overnight at 4 °C with a COX-2-specific primary antibody (Cayman Chemical, Ann Arbor, MI, USA, Cat. #160106, 1:1000) after citrate-based antigen unmasking, quenching of exogenous peroxidases with 3% hydrogen peroxide in methanol, and blocking with 10% normal goat serum (NGS; Vector Laboratories, Burlingame, CA, USA, Cat. #S-1000) in pH 7.5 PBS. A biotinylated anti-rabbit secondary antibody (Vector Laboratories, Cat. #BA-1000) was applied, followed by incubation with horseradish peroxidase (Thermo Fisher Scientific, Waltham, MA, USA, Cat. #434323) and developing using the Vectastain Elite DAB kit (Vector Laboratories, Cat. #SK-4100).

DBA staining was performed largely as previously described [11] by incubation with biotinylated *Dolichos biflorus* lectin (Vector Laboratories, B-1035-5, 1:250) overnight at 4°C after treatment with 1% hydrogen peroxide in pH 7.5 PBS and blocking with 10% NGS (Vector Laboratories, Cat. #S-1000) in pH 7.5 PBS. Horseradish peroxidase (Thermo Fisher Scientific, Cat. #434323) was applied, followed by developing using the Vectastain Elite DAB kit (Vector Laboratories, Cat. #SK-4100).

For immunofluorescence, frozen, OCT-embedded tissues were cut at 10 μm, mounted on slides, fixed in 4% (*vol/vol*) paraformaldehyde (Fisher Scientific, Cat. #04042-500), immersed in 0.3% hydrogen peroxide in methanol, and washed in 1/40 Triton-X 100 (Fisher Scientific, Cat. #BP151-500) before blocking with 10% normal goat serum (NGS; Vector Laboratories, Cat. #S-1000) in pH 7.5 PBS and incubating with an F4/80 primary antibody (BioLegend, San Diego, CA, USA, Cat. #123101) diluted in 10% NGS in PBS overnight at 4 °C. An anti-Rat Alexa Fluor 488 secondary antibody (Thermo Fisher Scientific, A-21208) was used before mounting with VECTASHIELD^®^ Antifade Mounting Medium with DAPI (Vector Laboratories, Cat. #H-1200). Imaging was performed with a Nikon epi-fluorescence microscope and NIS-Elements imaging software (Nikon Instruments, Melville, NY, USA). The percentage of F4/80-positive uterine cells was counted in representative stromal fields of approximately 350 cells per sample. The percentage of F4/80-positive spleen cells was counted in representative red pulp fields of approximately 500 cells per sample.

### 4.4. RNA Isolation and RT-qPCR

As previously described [24], total RNA was extracted from uterine tissues using the RNeasy Mini Kit for total RNA isolation (Qiagen, Valencia, CA, USA, Cat. #74106). Template cDNA was produced from 3 μg of total RNA using random hexamers and MMLV Reverse transcriptase (Thermo Fisher Scientific, Cat. # 28025013). cDNA levels were measured by real-time PCR SYBR green analysis using an Applied Biosystems StepOnePlus system according to the manufacturer’s instructions (Applied Biosystems, Foster City, CA, USA) using pre-validated primers (Table 1) and PowerUp SYBR Green Master Mix (Thermo Fisher Scientific, Cat. #A25742). The amplification conditions were 10 min at 95 °C, 40 cycles of 15 s at 95 °C followed by 1 min at 60 °C, then a melt curve phase consisting of 15 s at 95 °C, 1 min at 60 °C, and 15 s at 95 °C. Expression levels were normalized against ribosomal protein L7 (*Rpl7*) transcripts.

### 4.5. RNA-Sequencing

Libraries were prepared by the Van Andel Genomics Core from 500 ng of total RNA using the KAPA mRNA Hyperprep kit (v4.17) (Kapa Biosystems, Wilmington, MA, USA). RNA was sheared to 300–400 bp. Prior to PCR amplification, cDNA fragments were ligated to IDT Illumina UDI dual Indexed adapters (Illumina Inc, San Diego, CA, USA). Quality and quantity of the finished libraries were assessed using a combination of Agilent DNF-474 HS fragment kit (Agilent Technologies, Inc.) using a Fragment Analyzer (Agilent Technologies, Inc.), and QuantiFluor^®^ dsDNA System (Promega Corp., Madison, WI, USA). Individually indexed libraries were pooled and 50 bp, paired end sequencing was performed on an Illumina NovaSeq6000 sequencer using an S2, 100 bp sequencing kit (Illumina Inc., San Diego, CA, USA) to an average depth of 50M reads per sample. Base calling was conducted by Illumina RTA3 and output of NCS was demultiplexed and converted to FastQ format with Illumina Bcl2fastq v1.9.0.

### 4.6. RNA-Sequencing Analysis

Based on previously described methods [29], raw reads were trimmed with cutadapt [81] v1.18 and Trim Galore! (http://www.bioinformatics.babraham.ac.uk/projects/trim_galore/) v0.6.4_dev. Quality control analysis was performed using FastQC (https://www.bioinformatics.babraham.ac.uk/projects/fastqc/) v0.11.9. Trimmed reads were aligned to the GRCm38 genome assembly using STAR [82] v2.7.3a, and feature counting was performed using the command “--quantMode GeneCounts”. Output gene count files were constructed into an experimental read count matrix in R. Low count genes were filtered (1 count per sample on average) prior to DESeq2 [83,84] v1.30.1 count normalization and subsequent differential expression analysis. Calculated differential expression probabilities were corrected for multiple testing by independent hypothesis weighting (IHW [85] v 1.18.0) for downstream analysis, referred to as false discovery rate (FDR) for simplicity. Differentially expressed gene thresholds were set at FDR < 0.05. All reported instances of log2 (fold-change) data from RNA-seq were adjusted by apeglm v1.12.0 for LFC shrinkage [86]. Principal component analysis was calculated based on the top 500 genes by variance using DESeq2 and plotted with ggplot2 [87] v3.3.3. The RNA-seq heatmap was generated with ComplexHeatmap v2.6.2 [88] and circlize v0.4.12 [89] using scaled regularized-logarithm (rlog) counts for visualization. Differentially expressed genes were classified using QIAGEN Ingenuity Pathway Analysis (IPA) with results plotted with ggplot2.

### 4.7. Flow Cytometry

Uterine horns from GD3.5 mice were excised, spliced open longitudinally, and finely chopped using spring scissors. To prepare single cell suspensions, uterine tissue fragments were continuously digested in Enzyme buffer (Liberase™ TH Research Grade (0.09625 mg/mL, Millipore Sigma, Burlington, MA, USA, Cat. #LIBTH-RO), 100 U of DNaseI (Millipore Sigma, Cat. #AMPD1) in RPMI 1640 (Gibco, Grand Island, NY, USA, Cat. #11835-030) at 37 °C. To aid dissociation, samples were taken out every 20 min and pushed through 18 and 23 gauge needles. Once fragments were dissociated (after 60 to 90 min), samples were filtered through 70 um mesh, centrifuged at 400× *g* for 5 min, then resuspended in flow staining buffer (5% fetal bovine serum in PBS) to neutralize the Liberase™ TH enzyme. Red blood cells were lysed by resuspending the cell pellet in 1 mL of ACK (Ammonium-Chloride-Potassium) lysis buffer. To collect peritoneal fluid samples, 2–3 mL PBS was injected into the peritoneal cavity, the mouse was gently massaged, then as much fluid as possible was withdrawn. This was repeated if necessary to collect at least 2 mL of fluid. The resulting single cell suspensions (uterus and peritoneal lavage) were counted using Countess II (Thermo Fisher Scientific), resuspended in flow staining buffer and placed on ice.

To analyze viability, all cells were stained with Live/Dead Blue (Thermo Fisher Scientific, Cat. #L23105) on ice for 30 min, followed by Fc receptor block (BD Biosciences, East Rutherford, NJ, USA, Cat. #553141) on ice for 20 min. Antibodies against specific mouse antigens were utilized for cell-surface staining (Table 2). Between 2 – 6 × 10^5^ cells were stained using a cocktail of cell-surface antibodies for 30 min on ice, washed three times in flow staining buffer, and fixed in 1% paraformaldehyde prior to analysis. All samples were analyzed using spectral flow cytometry, Cytek^®^ Aurora (Cytek Biosciences, Fremont, CA, USA) at Michigan State University Flow Cytometry Core. The flow cytometric analysis software Kaluza (Beckman Coulter, Indianapolis, IN, USA) was used for analysis and generation of gating strategy.

### 4.8. Statistical Analysis

To assess statistical significance of parametric data, we used Student’s t-test. For non-parametric data, we used the Mann–Whitney U test. All statistical tests were two-tailed when applicable, and a value of *p* < 0.05 was considered statistically significant. Statistical analyses were performed using the Instat 3 package from GraphPad (San Diego, CA, USA). Statistical test results (*p*-values) are presented with the results in the text and symbolically in the figures, with explanations in figure legends. The value of n for each experiment, representing number of animals unless noted as number of implantation sites, is reported in the appropriate figure legend.

### 4.9. Data and Code Availability

The data that support the findings of this study are available from the article, Appendix A, and the publicly accessible GEO database under series GSE196489.

## Figures and Tables

**Figure 1 ijms-23-06067-f001:**
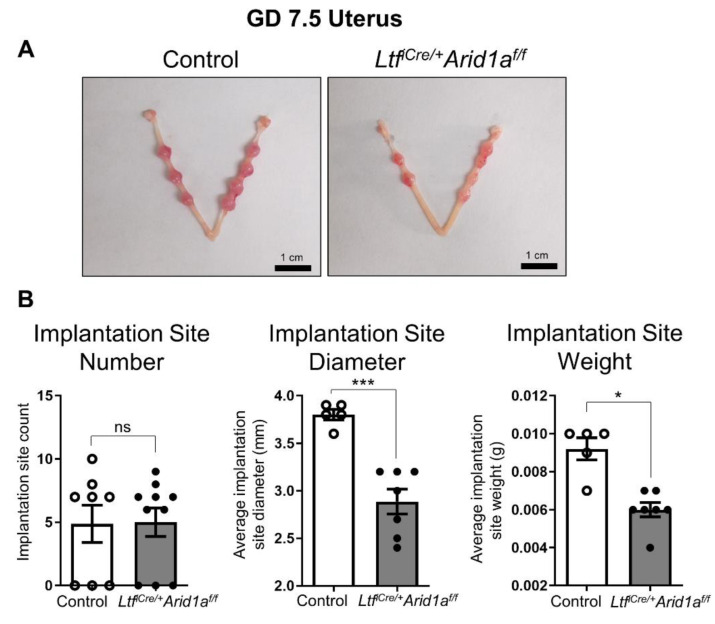
Deletion of endometrial epithelial *Arid1a* in mice causes diminished implantation site size and uNK cell numbers at GD 7.5. (**A**) Implantation sites were grossly visible in both control (*n* = 8) and *Ltf^iCre/+^Arid1a^f/f^* (*n* = 10) uteri at GD 7.5, appearing smaller in *Ltf^iCre/+^Arid1a^f/f^* uteri. (**B**) Implantation site number (left) as counted based on gross morphology was similar between control (*n* = 8, empty bar) and *Ltf^iCre/+^Arid1a^f/f^* (*n* = 10, grey bar) uteri at GD 7.5, but the average implantation site diameter (middle) and weight (right) were significantly decreased in *Ltf^iCre/+^Arid1a^f/f^* (*n* = 7, grey bar) uteri compared to controls (*n* = 5, empty bar). The graphs represent the mean ± SEM. * *p* < 0.05; ***, *p* < 0.001; ns, *p* > 0.05. Abbreviations: *Arid1a*, AT-rich interaction domain 1A; GD, gestation day; *Ltf*, lactoferrin; uNK, uterine natural killer.

**Figure 2 ijms-23-06067-f002:**
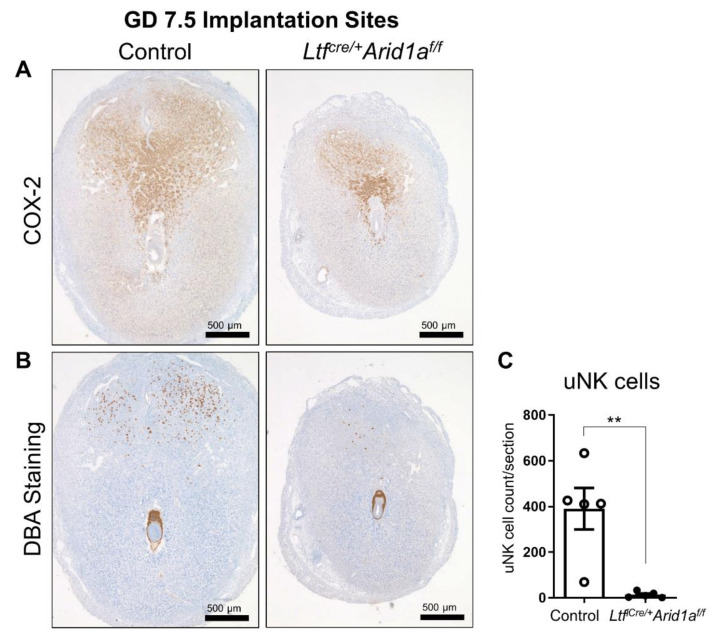
Deletion of endometrial epithelial *Arid1a* in mice causes diminished decidual area and decreased uNK cell numbers in GD 7.5 implantation sites. (**A**) Representative images of COX-2 IHC in control and *Ltf^iCre/+^Arid1a^f/f^* mouse uteri at GD 7.5 (*n* = 5 IS/genotype). (**B**) Representative images of DBA lectin staining for uNK cells in control and *Ltf^iCre/+^Arid1a^f/f^* mouse uteri at GD 7.5 (*n* = 5 IS/genotype). (**C**) Quantification of the average number of DBA-stained uNK cells counted per tissue section in control (*n* = 5 IS, empty bar) and *Ltf^iCre/+^Arid1a^f/f^* (*n* = 5 IS, grey bar) uteri. The graph represents the mean ± SEM. **, *p* < 0.01. Abbreviations: *Arid1a*, AT-rich interaction domain 1A; COX-2, cyclooxygenase-2; DBA, *Dolichos biflorus*; GD, gestation day; IS, implantation site; *Ltf*, lactoferrin; uNK, uterine natural killer.

**Figure 3 ijms-23-06067-f003:**
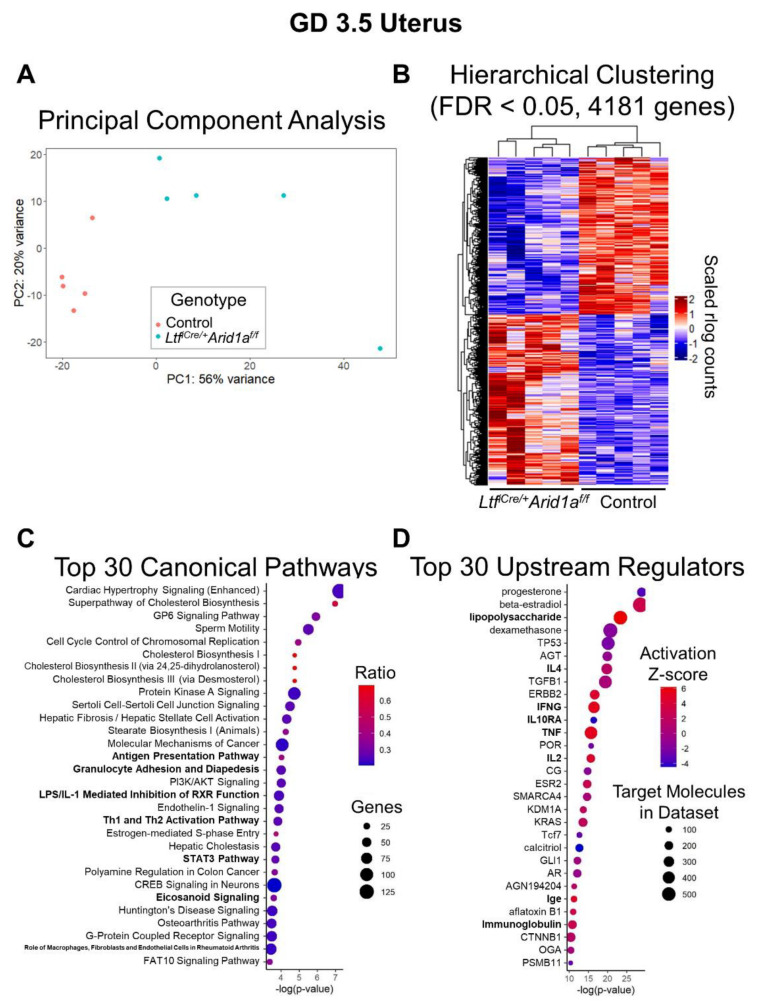
RNA-sequencing analysis of *Ltf^iCre/+^Arid1a^f/f^* uteri at GD 3.5 reveals large scale transcriptome dysregulation. (**A**) RNA-sequencing was performed on GD 3.5 control and *Ltf^iCre/+^Arid1a^f/f^* uterine RNA samples (*n* = 5/genotype). The PCA plot graphically shows that the overall gene expression patterns are distinct between groups. The plot was created using DESeq2 and ggplot2 in Rstudio. (**B**) Differential expression analysis with DESeq2 identified 4181 (2007 decreased, 2174 increased) significantly differentially expressed genes (DEGs) meeting the threshold of FDR < 0.05, corrected for multiple testing by independent hypothesis weighting. Hierarchical cluster analysis clearly distinguished the two groups based on gene expression patterns. The plot was created using ComplexHeatmap in Rstudio. (**C**) Ingenuity Pathway Analysis of the 4181 genes differentially expressed between control and *Ltf^iCre/+^Arid1a^f/f^* uteri (FDR < 0.05) identified 194 significantly enriched canonical pathways (*p* < 0.05). The plot shows the top 30 pathways from this list based on their corresponding *p*-values and also displays ratios (genes in current set/total genes in pathway) and number of genes from each pathway dysregulated in *Ltf^iCre/+^Arid1a^f/f^* uteri. (**D**) Ingenuity Pathway Analysis of the 4181 genes differentially expressed between control and *Ltf^iCre/+^Arid1a^f/f^* uteri (FDR < 0.05) identified 1526 significantly enriched upstream regulators (*p* < 0.05). The plot shows the top 30 upstream regulators from this list based on their corresponding p-values and also displays activation z-score and number of target molecules in the dataset. The plots were created with ggplot2 in Rstudio. Abbreviations: *Arid1a*, AT-rich interaction domain 1A; FDR, false discovery rate; GD, gestation day; *Ltf*, lactoferrin; PCA, principal component analysis.

**Figure 4 ijms-23-06067-f004:**
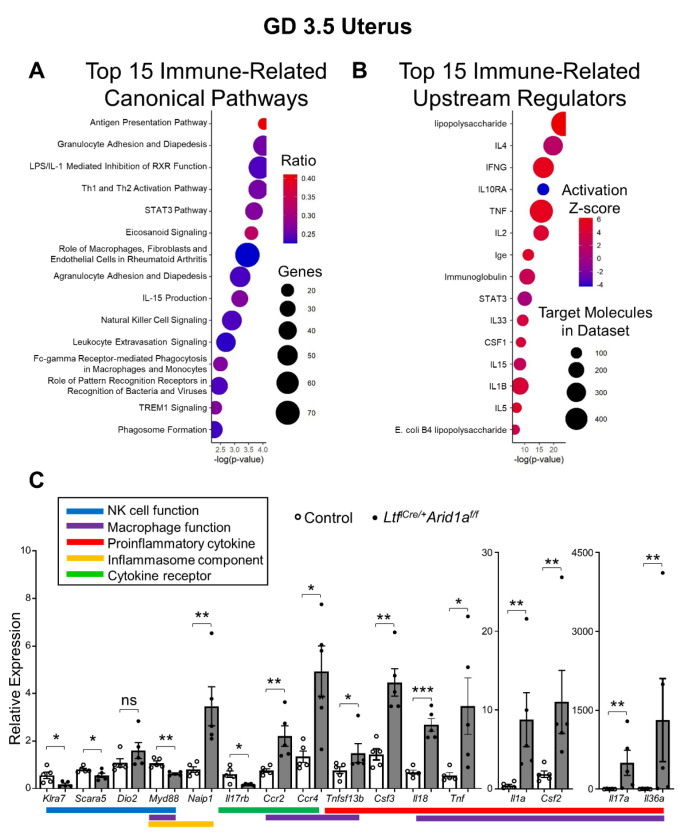
*Ltf^iCre/+^Arid1a^f/f^* uteri exhibit immune-related gene expression changes at GD 3.5. (**A**) Ingenuity Pathway Analysis of the 4181 genes differentially expressed between *Arid1a^f/f^* and *Ltf^iCre/+^Arid1a^f/f^* uteri (FDR < 0.05) identified 194 significantly enriched canonical pathways (*p* < 0.05). The plot shows the top 15 immune-related pathways from this list based on their corresponding *p*-values and also displays ratios (genes in current set/total genes in pathway) and number of genes from each pathway dysregulated in *Ltf^iCre/+^Arid1a^f/f^* uteri. (**B**) Ingenuity Pathway Analysis of the 4181 genes differentially expressed between *Arid1a^f/f^* and *Ltf^iCre/+^Arid1a^f/f^* uteri (FDR < 0.05) identified 1526 significantly enriched upstream regulators (*p* < 0.05). The plot shows the top 15 immune-related upstream regulators from this list based on their corresponding *p*-values and also displays activation z-score and number of target molecules in the dataset. The plots were created with ggplot2 in Rstudio. (**C**) Relative expression levels of the mRNA from each gene were normalized to *Rpl7* in whole uterine RNA preparations from control (*n* = 5, empty bar and empty dot) and *Ltf^iCre/+^Arid1a^f/f^* (*n* = 5, grey bar and filled dot) uteri at GD3.5 determined with RT-qPCR. The graphs represent the mean ± SEM. *, *p* < 0.05; **, *p* < 0.01; ***, *p* < 0.001; ns, *p* > 0.05. Abbreviations: *Arid1a*, AT-rich interaction domain 1A; *Ccr2*, C–C motif chemokine receptor 2; *Ccr4*, C–C motif chemokine receptor 4; *Csf2*, colony-stimulating factor 2; *Csf3*, colony-stimulating factor 3; *Dio2*, iodothyronine deiodinase 2; FDR, false discovery rate; GD, gestation day; *Klra7*, killer cell lectin-like receptor 7; *Il1a*, interleukin 1 alpha; *Il17a*, interleukin-17A; *Il17rb*, interleukin-17 receptor B; *Il18*, interleukin-18; *Il36a*, interleukin-36 alpha; *Ltf*, lactoferrin; *Myd88*, myeloid differentiation primary response 88; *Naip1*, NLR family, apoptosis inhibitory protein 1; NK, natural killer; PCA, principal component analysis; *Rpl7*, ribosomal protein L7; *Scara5*, scavenger receptor class A member 5; *Tnf*, tumor necrosis factor; *Tnfsf13b*, tumor necrosis factor ligand superfamily member 13B.

**Figure 5 ijms-23-06067-f005:**
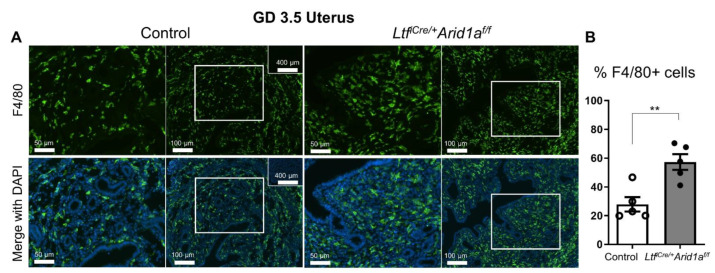
Uterine F4/80+ macrophage numbers are elevated in *Ltf^iCre/+^Arid1a^f/f^* mice at GD 3.5. (**A**) Representative images of F4/80 immunofluorescence (green) counterstained with DAPI (blue) in control (left) and *Ltf^iCre/+^Arid1a^f/f^* (right) mouse uterine sections at GD 3.5 (*n* = 5/genotype). For each group, the left image is higher magnification (scale bar = 50 µm) and the right image is a lower magnification region containing the zoomed region (indicated by large white rectangle; scale bar = 100 µm). The small insets in the upper right corner of the control images show no primary antibody negative controls (scale bar = 400 µm). (**B**) The percentage of F4/80-positive uterine cells was counted in representative stromal fields of approximately 350 cells per sample of control (*n* = 5, empty bar) and *Ltf^iCre/+^Arid1a^f/f^* (*n* = 5, grey bar) uteri at GD3.5. The graph represents the mean ± SEM. **, *p* < 0.01. Abbreviations: *Arid1a*, AT-rich interaction domain 1A; F4/80, EGF-like module-containing mucin-like hormone receptor-like 1; GD, gestation day; *Ltf*, lactoferrin.

**Figure 6 ijms-23-06067-f006:**
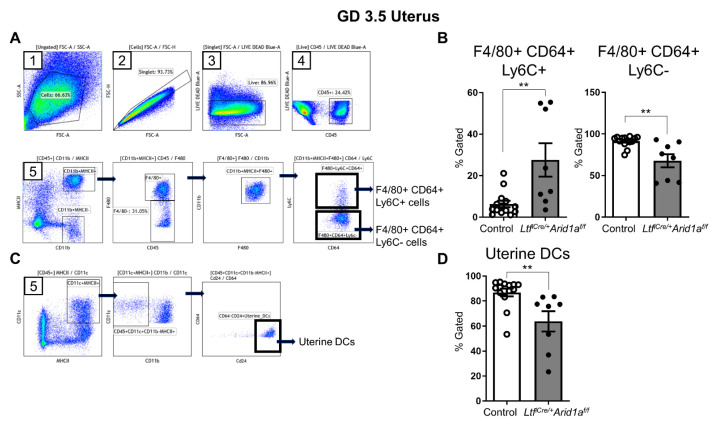
F4/80+ CD64+ Ly6C+ cells are increased, while uDCs are decreased in *Ltf^iCre/+^Arid1a^f/f^* uteri at GD 3.5. (**A**) The flow gating strategy to identify F4/80+ CD64+ Ly6C+ and F4/80+ CD64+ Ly6C-cells in the mouse uterus is shown. (**B**) The percentage Gated proportion of F4/80+ CD64+ Ly6C+ cells (left) and F4/80+ CD64+ Ly6C− (right) cells in control (*n* = 14, empty bar) and *Ltf^iCre/+^Arid1a^f/f^* (*n* = 8, grey bar) uteri at GD 3.5 is shown. (**C**) The flow gating strategy to identify uDCs in the uterus is shown. (**D**) The percentage Gated proportion of uDCs in control (*n* = 14, empty bar) and *Ltf^iCre/+^Arid1a^f/f^* (*n* = 8, grey bar) uteri at GD 3.5. The graphs represent the mean ± SEM. **, *p* < 0.01. Abbreviations: *Arid1a*, AT-rich interaction domain 1A; CD64, cluster of differentiation 64; DCs, dendritic cells; F4/80, EGF-like module-containing mucin-like hormone receptor-like 1; GD, gestation day; *Ltf*, lactoferrin; Ly6c, lymphocyte antigen 6C.

**Figure 7 ijms-23-06067-f007:**
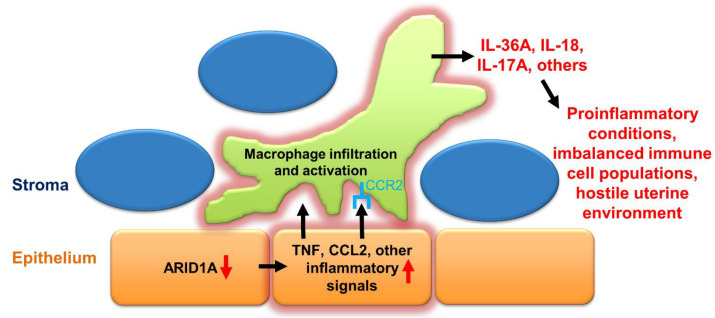
Proposed mechanism: Endometrial epithelial ARID1A loss leads to increased proinflammatory cytokine expression and macrophage-driven uterine inflammation during early pregnancy. Abbreviations: ARID1A, AT-rich interaction domain 1A; CCL2, monocyte chemoattractant protein-1; CCR2, C–C motif chemokine receptor 2; IL-17A, interleukin-17A; IL-18, interleukin-18; IL-36A, interleukin-36 alpha; TNF, tumor necrosis factor.

**Table 1 ijms-23-06067-t001:** SYBR green primers used for RT-qPCR.

Gene Name	Forward Primer (5′–3′)	Reverse Primer (5′–3′)	Source
*Klra7*	TCACAGCACACAGGTAGAGG	AGCTGGAAATCTGGCAGGTC	Self-designed
*Scara5*	AGGAGGGAAAGCCAGGTAGC	CCCCTAGCTTCCCATCATCA	[69]
*Dio2*	CCTCTTCCTGGCGCTCTATG	TTCAGGATTGGAGACGTGCA	[70]
*Myd88*	CCCACTCGCAGTTTGTTGGA	TAGGGGGTCATCAAGGGTGG	Self-designed
*Naip1*	CAGCCACCTAAAATAAGCTCTGG	GGACCCATGTTGGTCACTCC	[71]
*Il17rb*	CCATCCCTCCAGATGACAAC	TGCTCCTTCCTTGCCTCCAAGTTA	[72]
*Ccr2*	GACAAGCACTTAGACCAGGC	ACCTTCGGAACTTCTCTCCA	[73]
*Ccr4*	CCATTCTGGGGCTACTACGC	ACCAGGTACATCCATGAAACGA	[73]
*Tnfsf13b*	ACACTGCCCAACAATTCCTG	TCGTCTCCGTTGCGTGAAATC	[74]
*Csf3*	GCAGACACAGTGCCTAAGCCA	CATCCAGCTGAAGCAAGTCCA	[75]
*Il18*	GACTCTTGCGTCAACTTCAAGG	CAGGCTGTCTTTTGTCAACGA	[76]
*Tnf*	GCCTCCCTCTCATCAGTTCT	CACTTGGTGGTTTGCTACGA	[77]
*Il1a*	CCATCCAACCCAGATCAGCA	GTTTCTGGCAACTCCTTCAGC	[78]
*Csf2*	CCTGGGCATTGTGGTCTACAG	GGCATGTCATCCAGGAGGTT	[75]
*Il17a*	GGAGAGCTTCATCTGTGTCTCTG	TTGGCCTCAGTGTTTGGACA	[79]
*Il36a*	CTACAGCTTGGGGAAGGGAACATA	CCCTTTAGAGCAGACAGCGATGAA	[80]

**Table 2 ijms-23-06067-t002:** Antibodies used for flow cytometry.

Antibody	Conjugate	Clone	Company	Cat. #
CD45	PE Cy5	30-F11	BioLegend, San Diego, CA, USA	103110
CD11c	PE	N418	BioLegend, San Diego, CA, USA	117307
CD11b	Percp Cy5.5	M1/70	BioLegend, San Diego, CA, USA	101228
Ly6C	BV510	HK1.4	BioLegend, San Diego, CA, USA	128033
Ly6G	BV711	1A8	BioLegend, San Diego, CA, USA	127643
CD64	FITC	X54-5/7.1	BioLegend, San Diego, CA, USA	136316
CD24	APC	M1/69	BioLegend, San Diego, CA, USA	101813
F4/80	APC Cy7	BM8	BioLegend, San Diego, CA, USA	127117
MHCII	EF450	M5/114.15.2	Invitrogen, Waltham, MA, USA	48-5321-82
LD	Blue	-	Thermo Fisher, Waltham, MA, USA	L23105

## Data Availability

The data that support the findings of this study are available from the article, Appendix A, and the publicly accessible GEO database under series GSE196489.

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
