# Peer review of "Endometrial Epithelial ARID1A Is Required for Uterine Immune Homeostasis during Early Pregnancy"

_ijms, 2022, doi:10.3390/ijms23116067_

Round 1

Reviewer 1 Report

An excellent, well-researched and well-written article. There are minimal comments.

  1. Perhaps "preimplantation pregnant uteri" in abstract should be replaced with "preimplantation uteri", because the original combination of words is some kind of oxymoron.
  2. It is worth changing the caption to Figure 2: the general name refers only to 2B, but not 2A
  3. There is no S1 table in Supplemental Materials

Author Response

C1. An excellent, well-researched and well-written article. There are minimal comments.

A1. We thank the reviewer for the kind words and constructive comments, which have helped improve the quality of the manuscript.

C2. Perhaps "preimplantation pregnant uteri" in abstract should be replaced with "preimplantation uteri", because the original combination of words is some kind of oxymoron.

A2. Thank you for this note; we have changed the wording as requested.

C3. It is worth changing the caption to Figure 2: the general name refers only to 2B, but not 2A

A3. This point is well taken. We have replaced the caption title with “Deletion of endometrial epithelial Arid1a in mice causes diminished decidual area and decreased uNK cell numbers in GD 7.5 implantation sites.”

C4. There is no S1 table in Supplemental Materials

A4. Thank you for catching this. Table S1 has been added as a separate .xlsx file.

Reviewer 2 Report

The current study attempts to evaluate the effect of a well-described gene, ARID1A in immune homeostasis during early pregnancy.

It is a well-designed experimental setup, with a large set of different analytical techniques, that allow authors to study markers in the tissue at the protein level by IHC, flow cytometry, and also at mRNA level with qPCR.

Comments: 

· One of the limitations of the study, which is included by the authors at the end of the discussion section, is the lack of selection, in the current experiment,  of the different parts of the endometrial tissue. This analysis of the bulk tissue, makes me suspect that some results should be carefully addressed and the final conclusions could be influenced by this non-selected cellular type. How do you consider that it could be addressed these plausible differences in tissues both at protein and mRNA levels? Did you consider the use of differential markers to address/quantify the percentage of every different layer of the endometrium that contains the samples before analysis of proteins or mRNA?

· This doubt is also extensible to the flow cytometry analysis, whose digestion is, as far as I understand, without selection of the endometrial layers.

· Figure 7 indicates epithelium and I consider that, after my two previous comments, it is quite risky to indicate only this cellular type.

· Finally, murine models in reproduction events have been a matter of debate lately (Cunha GR, Sinclair A, Ricke WA, Robboy SJ, Cao M, Baskin LS. Reproductive tract biology: Of mice and men. Differentiation. 2019 Nov-Dec;110:49-63. doi: 10.1016/j.diff.2019.07.004. Epub 2019 Sep 6. PMID: 31622789; PMCID: PMC7339118). So, how could you hypothesize that such differences in an organism should be biasing your results? 

Author Response

C1. The current study attempts to evaluate the effect of a well-described gene, ARID1A in immune homeostasis during early pregnancy.

It is a well-designed experimental setup, with a large set of different analytical techniques, that allow authors to study markers in the tissue at the protein level by IHC, flow cytometry, and also at mRNA level with qPCR.

A1. We thank the reviewer for the kind comments and insightful questions.

C2. One of the limitations of the study, which is included by the authors at the end of the discussion section, is the lack of selection, in the current experiment,  of the different parts of the endometrial tissue. This analysis of the bulk tissue, makes me suspect that some results should be carefully addressed and the final conclusions could be influenced by this non-selected cellular type. How do you consider that it could be addressed these plausible differences in tissues both at protein and mRNA levels? Did you consider the use of differential markers to address/quantify the percentage of every different layer of the endometrium that contains the samples before analysis of proteins or mRNA?

A2. This is indeed a limitation of our study as we mentioned in the Discussion. Yes, we considered the possibility of isolating epithelial and stromal cell layers from the mouse uterus for our RNA sequencing and RT-qPCR analyses; however, this is technically challenging and requires pooling multiple samples. In order to have enough biologically distinct sample tissue for each assay, we used bulk tissues. However, our immunostaining (Figure 5) did allow us to differentiate between cell layers with regard to the macrophage population based on the surface marker protein F4/80.

C3. This doubt is also extensible to the flow cytometry analysis, whose digestion is, as far as I understand, without selection of the endometrial layers.

A3. Yes, this is also true for the flow cytometry analysis. The small size of the mouse uterus makes selection of the individual layers for flow cytometry very challenging. However, as explained above, our immunostaining revealed that the F4/80+ population was located almost entirely within the endometrial stroma and myometrium rather than the epithelium. This finding suggests that the F4/80+ populations identified by flow cytometry (Figure 6) were located in the stromal and myometrial compartments before tissue digestion and not the epithelium.

C4. Figure 7 indicates epithelium and I consider that, after my two previous comments, it is quite risky to indicate only this cellular type.

A4. In addition to the epithelium, we also show the involvement of the stroma in this figure. While it is true that our RNA and flow cytometry analyses in this manuscript were done on bulk tissues, immunostaining showed that the F4/80+ macrophages were located in the stroma. Furthermore, the proposed mechanism in the figure is based on data from published literature in addition to the current study. For example, our previous study [PMID: 33222288] demonstrated that ARID1A expression is lost in endometrial epithelium but not the stroma of LtfiCre/+Arid1af/f mice. The epithelial-specific secretion of TNF and the transcriptional regulation of TNF signalling factors and CCL2 by ARID1A as shown in the figure have been reported by other research groups [PMID: 8671447, 31391455]. Therefore, while some assays in our present study did not distinguish between the epithelium and stroma, the information represented in the figure was derived from other studies that did distinguish between the cell types and were cited in the text.

C5. Finally, murine models in reproduction events have been a matter of debate lately (Cunha GR, Sinclair A, Ricke WA, Robboy SJ, Cao M, Baskin LS. Reproductive tract biology: Of mice and men. Differentiation. 2019 Nov-Dec;110:49-63. doi: 10.1016/j.diff.2019.07.004. Epub 2019 Sep 6. PMID: 31622789; PMCID: PMC7339118). So, how could you hypothesize that such differences in an organism should be biasing your results?

A5. This is an important point. One difference between mice and humans that could impact our findings is the differential uterine hormone regulation of the murine estrous cycle and human menstrual cycle. Since hormones can affect immune cell populations, this may limit the application of our findings to humans. In light of that, we have added the following sentences to our discussion of the limitations of the study:

Finally, all the experiments in this study utilized mice as a model system. While mice share many aspects of uterine biology with humans, there are important differences of reproductive tract development and function between species that must be considered [PMID: 31622789]. For example, the differences in the hormonal environment between the human menstrual cycle and murine estrous cycle could impact the basal immune cell populations [PMID: 16485018, 31424502]. Therefore, our findings still need to be validated in human samples.”